# Production of Cellulose Pulp and Lignin from High-Density Apple Wood Waste by Preimpregnation-Assisted Soda Cooking

**DOI:** 10.3390/polym15071693

**Published:** 2023-03-29

**Authors:** Shuai Gao, Guoyu Tian, Yingjuan Fu, Zhaojiang Wang

**Affiliations:** State Key Laboratory of Biobased Material and Green Papermaking, Qilu University of Technology (Shandong Academy of Sciences), Jinan 250353, China

**Keywords:** apple wood, alkali preimpregnation, lignin, pulping, soda cooking

## Abstract

Apple wood waste (AWW), mainly tree trucks, is collectible lignocellulosic biomass from orchard rotation. The biorefinery of AWW is challenging because of the hard and dense structure. In the present work, chemical composition determination and microstructure observation was performed for the first time on AWW. Alkali-preimpregnation-assisted soda cooking (APSC) was developed to separate cellulose a pulp and lignin from AWW. APSC attained pulp yield of 34.2% at 23% NaOH, showing a 13.2% improvement compared to conventional soda cooking (SC). Fiber length analysis showed APSC-AWW pulp consisted mainly of medium and short fibers, which means blending with long-fibered pulp to enhance the physical strength of pulp sheets. A blend of APSC-AWW pulp and long-fibered pulp in the proportion of 80:20 attained comparable physical strength to hardwood kraft pulp. ASPC-AWW lignin was separated from spent liquor by acidification and then purified by dialysis desalination. The purified ASPC-AWW lignin showed a weight-average molecular weight of 4462 g/mol, similar to softwood kraft lignin but more uniform. Structural analysis revealed that ASPC-AWW lignin was composed of a syringyl unit (S), guaiacyl unit(G), and *p*-hydroxyphenyl unit (H), and an S unit was dominant with an S/G/H ratio of 74.5:18.2:7.3. It is believed the utilization of fruit tree wood waste as the feedstock of biorefinery is attractive to countries without sufficient forestry resources. Furthermore, the developed APSC is based on conventional SC, which ensures the feasibility of an industrial application.

## 1. Introduction

Apple (*Malus domestica*), a domesticated tree and fruit of the rose family (Rosaceae), is one of the most widely cultivated tree fruits. Apple trees generally thrive from latitude 30° to 60°, both north and south. Fruit orchard management requires annual pruning, which leaves abundant residues on the ground. Furthermore, when fruit production declines, the trees are dismantled to make room for new ones. Therefore, a considerable amount of apple wood waste (AWW) is produced every year from the pruned branches or orchard rotation [1].

AWW is used as firewood because the density is as high as 780 kg/m^3^, and the recoverable heat value of dried wood is 15 MJ/kg, making it burns hot but slowly [2]. Except for being used as firewood, AWW is mainly used to prepare activated carbon or charcoal. Chao Liu et al. prepared biochar as a microbial absorbent using apple wood through microwave-assisted catalytic pyrolysis [3]. Similarly, the removal of aqueous Cr(VI) was investigated using biochar prepared from apple wood [4]. Carbonized apple branches have been reported to help improve the utilization of nitrogen fertilizer [5]. Withouck et al. investigated the solvent extraction of antimicrobial and antioxidant compounds, specifically polyphenols and flavonoids from the bark and core of apple wood [6].

It is necessary to explore the valorization strategies of AWW for the production of higher-value products, especially for the tree trunk, which is woody biomass, and is easy to collect and transport. Biorefinery emphasizes the comprehensive utilization of chemical compositions of lignocellulosic feedstock and provides an alternative for turning waste biomass into value-added products [7]. The pulp mill has been the most extensive biorefinery facility that produces, besides the principal product pulp (mainly from cellulose), and various lignin-derived fractions [8,9]. Next to cellulose, lignin is the second most abundant biopolymer and the main source of aromatic structures on earth. The emergence of pulp mill-based biorefinery projects has brought about renewed interest in the last decade for lignin and its potential use in polymer materials [10]. For many reasons, the pulping industry is gradually shifting with new to utilize various low-cost, more efficient lignocellulosic feedstock, such as agricultural straws and woody waste from orchard management, especially in poor forest countries [11]. Biorefinery of the low-cost lignocellulosic biomass improves waste management and makes more value-added bio-products while creating both jobs and local economic development.

The key to AWW valorization in pulp mill-based biorefinery is to develop an effective cooking way of delignification. In this paper, the chemical compositions and microstructure of apple wood were studied prior to the exploration of cooking conditions. The conventional soda cooking was modified to overcome the hard and dense structure of AWW and therefore facilitate delignification. The physical strength of the cellulose pulp and the structural features of lignin were analyzed.

## 2. Materials and Methods

### 2.1. Materials

AWW was collected in an orchard in Shanxi province, China. The air-dried AWW consisted of tree trunk and branches. The tree trunk was manually cut into wood chips with a thickness of 0.5 cm and a length of 2 cm for cooking. Unbleached kraft pulp of *Pinus massoniana* (KP-P) was provided by Tiger Forest & Paper Group Co., Ltd. (Changsha, China). Unbleached kraft pulp of *Eucalyptus* and *Acacia Rachii* (KP-EA) was provided by Asia Symbol (Shandong) pulp and paper Co., Ltd. (Rizhao, China). Recycled pulp from old, corrugated container (RP-OCC) was made in laboratory. NaOH of analytical grade was purchased from Shanghai Aladdin Reagent Co., Ltd. (Shanghai, China).

### 2.2. Cooking of AWW to Isolate Pulp and Lignin

The conventional soda cooking (SC) of 40 g AWW was carried out using a stainless steel reactor with a volume of 1.5 L. Then cooking liquor of NaOH was added at different levels from 15% to 26% on weight basis of AWW. The ratio of AWW to cooking liquor (mass/volume) was 1:4. SC was kept at 160 °C for 4 h. To overcome the dense and hard structure of AWW, alkali-preimpregnation-assisted soda cooking (APSC) was proposed. As shown in Figure 1, AWW was soaked in spent liquor of soda cooking at 90 °C under atmospheric pressure for 90 min in the step of alkali-preimpregnation. The impregnated woodchips were then separated from spent liquor using a screen and then digested in fresh cooking liquor at 160 °C for 150 min. After digestion, pulp was separated from spent liquor by squeezing and water-washing. The washed pulp was disintegrated at 1.2% consistency for 3000 revolutions. The dispersed pulp slurry was screened using Somerville-type screening device with slit width of 0.15 mm according to TAPPI/ANSI T 275 sp-18. The yield of the screened pulp was calculated for evaluation of pulping performance. The weight of screen rejects, mostly botanical knots and insufficiently cooked woodchips, was also recorded. Lignin in spent liquor was isolated by acidification and dialysis processes [12]. Spent liquor was adjusted to pH 2.5 using sulfuric acid for lignin precipitation. The lignin precipitate was purified using a dialysis tube with a molecular weight cut-off of 500Da by removing mineral salts. The purified lignin was lyophilized for further analysis.

### 2.3. Forming Handsheets for Physical Tests

Handsheets of 60 g/m^2^ were prepared using the screened pulp of AWW for physical tests according to TAPPI/ANSI T 205 sp-18. Pulp beating was performed prior to handsheets forming according to TAPPI/ANSI T 248 sp-15 using a PFI mill to Canadian Standard Freeness (CSF) of 325 ± 20 mL. The physical properties of pulp handsheets were determined according to TAPPI/T220 sp-16, including tensile strength (TAPPI T494), burst strength (TAPPI T403), and tearing resistance (TAPPI T414). Each data point is the average of at least three replicates.

### 2.4. Pulp Fiber Length Classification

The classification of pulp into a number of fiber fractions using screens of 15, 30, 50, 100, and 200 meshes was conducted using Bauer-McNett classifier according to T233 cm-15. Weight percents retained on each screen and weight percent that passed through the last screen were calculated.

### 2.5. Structural Carbohydrate and Lignin Determination

The structural carbohydrates and lignin of AWW were determined according to technical report NREL/TP-510-42618, as previous studies described [13].

### 2.6. Lignin Molecular Weight Determination

Size exclusion chromatography (SEC) was used to determine the molecular weight distribution of lignin on a Waters e2695 system with three tandem gel columns (Phenolgel^TM^ 10,000Å, 500 Å, 50Å, 300 mm × 7.8 mm) as previous studies described [14]. Lignin was acetylated prior to SEC to make it fully soluble in eluent tetrahydrofuran (THF). GPC conditions: eluent THF flow rate 1.0 mL/min, column temperature 30 °C, UV detector at 280 nm. Polystyrene was used as the standard sample.

### 2.7. Lignin Structure Characterization

The 2D ^1^H-^13^C heteronuclear single quantum coherence nuclear magnetic resonance (2D ^1^H-^13^C HSQC NMR) was used for structural analysis of lignin. Lignin of 60 mg was dissolved in 0.6 mL of deuterated dimethyl sulfoxide (DMSO-d6) and analyzed by NMR using Bruker AVANCE II 400M. The number of ^13^C scans was 20,000, with collection time 0.4 s and relaxation delay time 1.5 s. The ^1^H scans were eight times with a width of 400 MHz, collection time 2 s, and relaxation time 3 s. The chemical shift of the solvent DMSO-d6 is 39.5 and 2.5 ppm. The 2D NMR spectrum was processed with Top-Spin 3.5b.

### 2.8. FTIR

APSC-AWW lignin and kraft lignin were analyzed by Fourier transforms infrared spectrometer. The samples were taken for analysis by FTIR (Bruker Tensor 37, Germany, ATR accessory). The scan range was from 400 cm^−1^ to 4000 cm^−1^, the resolution was set to 4 cm^−1^, and each sample was scanned 64 times.

### 2.9. TGA

APSC-AWW lignin and kraft lignin were heated from room temperature to 800 °C at a rate of 10 °C min^−1^ under air atmosphere by using Rigaku Thermo plus EVO2 TG 8121 system for thermogravimetric analysis (TGA).

### 2.10. Microstructure Detection of AWW and Pine Wood

Microcomputerized tomography (micro-CT) was used to obtain cross-sectional images of untreated AWW and pine wood at a resolution of 300 nm. Scanning electron microscope (SEM) was used to obtain surface topography images of palm fiber with a Hitachi Regulus 8220 SEM (Hitachi, Tokyo, Japan) that operated at an acceleration voltage of 5 kV.

## 3. Results and Discussions

### 3.1. Chemical Constituents, Density, and Microstructure of AWW

Cellulose, hemicelluloses, and lignin make up a major portion of the biomass. The chemical compositions of structural carbohydrates (cellulose and hemicelluloses) of AWW were determined according to the technical report NREL/TP-510-42618 issued by the U.S. Department of Energy. AWW is composed of 33.3% glucan, 16.8% xylan, 9.7% mannan, 2.2% galactan, and 0.5% arabinan. The lignin content of AWW was measured to be 29.1%, consisting of 23.7% acid-insoluble lignin and 5.4% acid-soluble lignin. To our knowledge, this is the first report on the chemical constituents of AWW. The glucan content of AWW is much lower than wood species conventionally used in pulping industry, e.g., 39.4–40.1% of poplar woods [15], 39.1% of eucalyptus [16], and 42.5% of lodgepole pine [17]. Worth mentioning is the mannan content of 9.7% in AWW, which is much higher than 1.4–3.9% in poplar woods [15]. The acid-insoluble lignin in AWW was 23.7%, in the same levels as 20.2–25.2% in poplar woods [15], much lower than 27.1% in lodgepole pine [17]. The ash content in AWW was determined to be 0.4%. It is well-acknowledged that a high level of ash leads to a buildup of ash and high rates of corrosion of pulping equipment, e.g., super-heater corrosion and plugging of the recovery boiler. The low ash content of AWW is a distinct advantage for application as pulping feedstock.

Figure 2 shows the density of the air-dried seasoned AWW, as well as the densities of the most common softwood and hardwood species used in the pulp industry. These data were collected from the website Engineering ToolBox. It is remarkable that the density of AWW is as high as 780 kg/m^3^, much higher than any other wood species listed in Figure 2. More accurately, the density of AWW is almost twice of aspen wood and 47% higher than Douglas fir. The high density of AWW means a dense structure, which is unfavorable to chemical penetration and delignification during the pulping process.

The microstructure of the transverse section of AWW and pine wood were obtained by micro-CT (Figure 3). To our knowledge, this is the first report about the microstructure of AWW. AWW shows some vessels. The lumen of the parenchyma cell of AWW is in the shape circle, while lumen of the parenchyma cell of pine wood is in the shape of a rectangle. The wall of the parenchyma cell of AWW is much thicker than pine wood, which explains the higher density of AWW and the higher lignin content [18]. Moreover, pine wood exhibits a loosened cytoskeleton and regular cell wall morphology. AWW exhibits a dense and hard structure.

### 3.2. Pulping of AWW by SC and APSC

Pulping is the process where fibers are separated from lignocellulosic biomass and treated to produce pulp. The separation of virgin lignocellulose into individual fibers involves mechanical, semi-chemical, or chemical treatments. The choice of pulping method depends on the feedstock properties and the quality requirements of the end product. Considering the dense and hard features of AWW, chemical pulping, namely, SC and APSC were conducted. Figure 4 shows the pulping performance of SC and APSC in terms of the screened pulp yield and the rejects yield. Rejects consisted mostly of botanical knots and insufficiently cooked woodchips. From Figure 4, the rejects yield of APSC was lower than that of SC at the same level of NaOH dosage, which is ascribed to the enhanced chemical permeation and delignification during the preimpregnation process. The preimpregnation-induced improvement of delignification was more profound at low levels of NaOH dosage, which had been confirmed by the lower Kappa numbers of APSC pulps than SC pulps in Figure 5. The rejects yield reduced to 0.7% at APSC-26, indicating that AWW had been almost entirely disintegrated into individual fibers. The positive effect of preimpregnation was also evidenced by the higher yields of the screened pulp of APSC than SC. From Figure 4, the difference in the screened pulp yields between SC and APAS was very significant at low NaOH dosage and gradually reduced with the increase in NaOH dosage. From Figure 4 and Figure 5, the screened pulp of APSC-26 showed a yield of 35.5% and a Kappa number of 16.2. It has been reported that the presence of lignin is good for the mechanical properties of a hot-pressed paper due to the lignin interdiffusion [19]. The high Kappa number pulps obtained at lower alkali dosages might be useful for the production of hot-pressed paper webs.

Fibers of the screened APSC-AWW pulp were classified by length into a number of fractions (P_>16_, P_16−30_, P_30−50_, P_50−100_, P_100−200_, P_<200_) using various screens in the Bauer-McNett classifier. Fiber classification of KP-P, KP-EA, and RP-OCC was also conducted for comparison. The weight percent of each fiber fraction was calculated and listed in Table 1. The top three fractions of APSC-AWW were P_30−50_, P_50−100_, and P_100−200_, with a sum of 83.7%, suggesting APSC-AWW mainly consists of medium and short fibers. KP-P is a widely used long-fibered pulp, which is confirmed by the sum of weight percent of 67.9% of P_>16_ and P_16−30_. KP-EA is a commercial pulp consisting of mainly medium fibers as indicated by the sum of weight percent 84.5% of P_30−50_ and P_50−100_. RP-OCC is a mixture of long fibers, medium fibers, and short fibers. It is worth pointing out that PR-OCC contains a great number of fines, which are generated from the dried fibers many times during recycling [20].

### 3.3. Physical Strength of APSC-AWW Pulp

Table 2 shows the physical properties of handsheets prepared using various pulps. The tensile index of APSC-AWW was 53.7 N·m/g, much higher than 43.6 N·m/g of recycled pulp RP-OCC and 24.9 N·m/g of palm fiber pulp SC-PF and comparable to 56.4 N·m/g of hardwood pulp KP-EA and 55.9 N·m/g of corn straw pulp SC-CS, but much lower than 75.8 N·m/g softwood pulp KP-P and 80.4 N·m/g of wheat straw pulps SC-WS. The tear index of APSC-AWW is 3.1 kPa·m^2^/g, lower than any other virgin pulps listed in Table 2. The low tear index of APSC-AWW is possibly ascribed to the lack of long fibers, as discussed above. The burst index of APSC-AWW is 3.2 mN·m^2^/g, higher than 2.3 kPa·m^2^/g of RP-OCC and 1.9 kPa·m^2^/g of SC-PF but lower than other virgin pulps listed in Table 2 Long fibered pulp, namely, the commercial pulp of KP-P, was added to APSC-AWW to improve the physical strength. Various APSC-AWW/KP-P blends in the proportion of 95:5, 90:10, and 80:20 were prepared, and then the handsheets properties of these blends were estimated, as shown in Table 2. As expected, the physical properties of APSC-AWW were greatly improved by the addition of long-fibered pulp KP-P. The tear index and burst index of APSC-AWW/KP-P_(80/20)_ reached 6.8 mN·m^2^/g and 4.3 kPa·m^2^/g, respectively, and comparable to the physical strength of hardwood pulp KP-EA.

### 3.4. Structural Features of APSC-AWW Lignin

APSC-AWW lignin was separated from spent liquor of pulping by acid precipitation and then purified by dialysis. The weight-average molecular weight (Mw), number-average molecular weight (Mn), and polydispersity index (PDI, Mw/Mn) of the purified APSC-AWW lignin were analyzed by SEC. In Table 3, molecular weights of softwood kraft lignin (Indulin kraft), soda lignin of mixed wheat straw and Sarkanda grass (Soda-P1000), and soda lignin from palm fiber (Soda-Palm) were cited for comparison. It is well known that the choice of eluent, column type, calibration standard, and lignin pretreatment all influence the molecular weight determination by SEC. For reliable comparison, molecular weights of lignin samples listed in Table 3 were determined under identical SEC conditions. The M_W_ of APSC-AWW lignin showed a value of 4462 g/mol, similar to the Mw of Indulin kraft and much higher than the Mw of Soda-P100 and Soda-Palm. APSC-AWW lignin showed a more uniform molecular weight distribution than Indulin kraft and Soda-P100, as indicated by the lower value of PDI. Lignin with higher Mw and lower PDI is preferred for dye dispersant production due to its outstanding interface performance and thermostability [24].

The 2D ^1^H-^13^C HSQC NMR was employed to investigate the structural information of APSC-AWW lignin. The spectrum of the side chain (δ_C_/δ_H_ 50–90/2.5–6.0) and the aromatic (δ_C_/δ_H_ 100–135/5.5–8.5) regions are shown in Figure 6. A small quantity of aryl ether bonds (A: β-O-4) was observed from the C_α_-H_α_ correlations at δ_C_/δ_H_ 71.7/4.86 with an abundance of 4.9 in every 100 aromatic units. The C_γ_-H_γ_ correlations in phenylcoumarane substructures (B: β-5,α-O-4) were observed at δ_C_/δ_H_ 62.5/3.73. The strong signals of C_α_-H_α_ correlations at δ_C_/δ_H_ 84.8/4.65, C_β_-H_β_ correlations at δ_C_/δ_H_ 53.5/3.06, and C_γ_-H_γ_ correlations at δ_C_/δ_H_ 71.0/3.82,4.18 indicated the presence of resinol substructures (C: β-β, α-O-γ, and γ-O-α). In the aromatic region, syringyl unit (S), guaiacyl unit (G), and *p*-hydroxyphenyl unit (H) were observed from the S_2,6_ at δ_C_/δ_H_ 103.8/6.71, G_2_ at δ_C_/δ_H_ 110.9/6.98, and H_2,6_ at δ_C_/δ_H_ 127.9/7.19, respectively. The calculated S/G/H ratio was 74.5:18.2:7.3 for APSC-AWW lignin, indicating the S unit is the dominant substructure. Generally speaking, the G unit is dominant in softwood lignin, which is the reason why sodium sulfide is needed for softwood delignification during the kraft pulping process.

Figure 7 shows the FTIR spectrum of APSC-AWW lignin and Indulin kraft lignin. Lignin has an alcoholic hydroxyl group and a phenolic hydroxyl group. The absorbance of the hydroxyl group at 3400 cm^−1^ of kraft lignin is stronger than APSC-AWW lignin, which suggests more cleavage of aryl ether bonds during kraft pulping. This is confirmed again by the lower absorbance of ether linkage at 1120 cm^−1^ for the kraft lignin than APSC-AWW lignin. Demethoxylation occurs during kraft pulping, which explains the lower absorbance of C-O stretching vibration at 1230–1220 cm^−1^ in the case of kraft lignin than APSC-AWW lignin.

Figure 8 shows the pyrolysis characteristics of APSC-AWW lignin and Indulin kraft lignin. Thermogravimetric analysis was conducted at a heating rate of 10 °C/min. The degradation of lignin took place in a temperature range from 200 °C to 800 °C. APSC-AWW lignin shows a faster thermal decomposition than kraft lignin. Temperatures of maximum degradation rate were 342 °C and 350 °C for APSC-AWW lignin and kraft lignin, respectively. The char yield of APSC-AWW lignin is 49%, lower than that of kraft lignin at 41%.

## 4. Conclusions

Many kinds of wood from fruit trees, such as apple wood, peach wood, and orange tree wood, have a dense and hard structure that is conventionally used as solid fuel. In this work, chemical composition determination and microstructure observation were conducted for the first time for apple wood. APSC process was developed to separate cellulose fibers and lignin. Although pulp from fruit trees consists mainly of medium and short fibers, blending with long-fibered pulp is feasible and practical to produce paper with excellent physical strength. APSC-AWW lignin showed similar molecular weight and thermal decomposition behavior to softwood kraft lignin. This biorefinery approach generates value-added products and, more importantly, avoid green gas emission. We believe the utilization of fruit tree wood as the feedstock of the pulp industry is attractive to countries without sufficient forestry resources. Furthermore, the developed APSC is based on conventional SC, which ensures the feasibility of an industrial application.

## Figures and Tables

**Figure 1 polymers-15-01693-f001:**
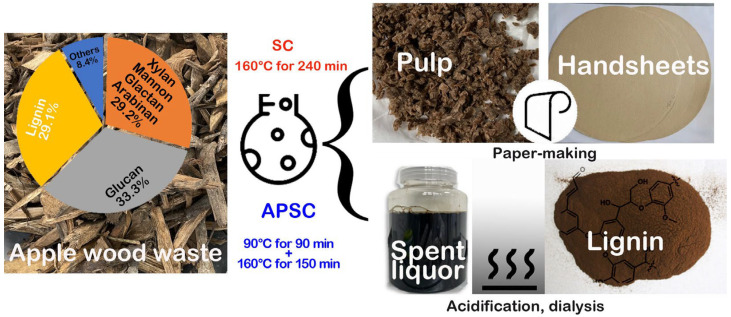
Flow diagram of the isolation of cellulose pulp and lignin form AWW by SC and APSC.

**Figure 2 polymers-15-01693-f002:**
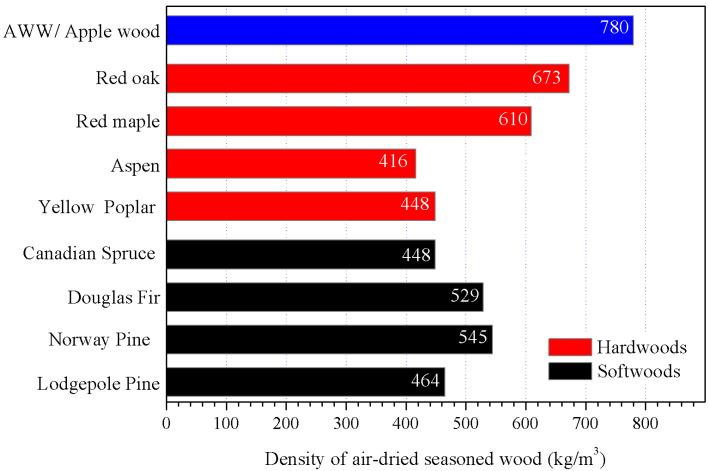
Comparison of densities of air-dried seasoned woods. Data are available online: https://www.engineeringtoolbox.com/weigt-wood-d_821.html (accessed on 30 January 2023).

**Figure 3 polymers-15-01693-f003:**
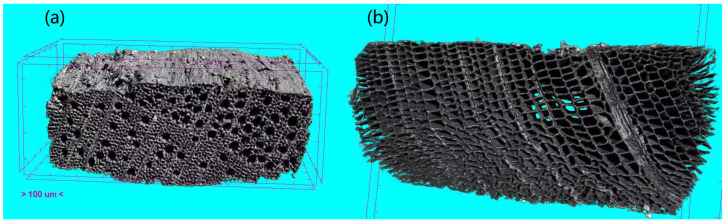
The transverse section images of AWW (**a**) and pine wood (**b**) obtained by microcomputerized tomography.

**Figure 4 polymers-15-01693-f004:**
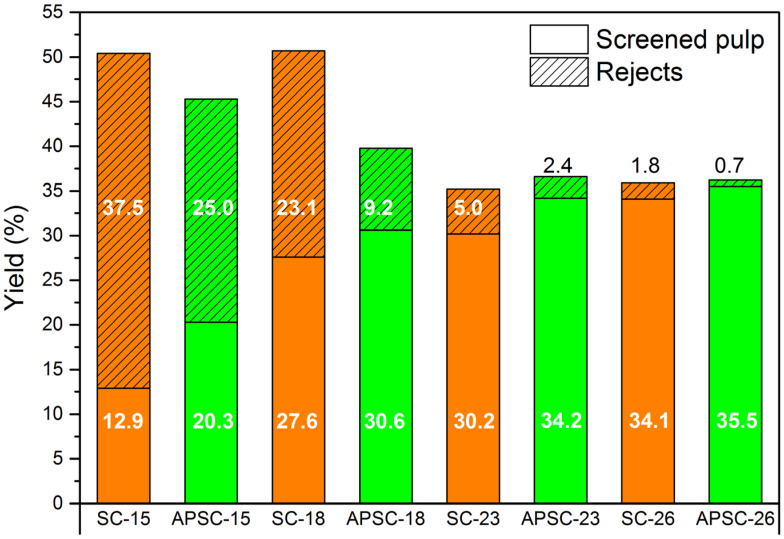
Pulping performance of SC and APSC in terms of yield of screened pulp and yield of rejects at different NaOH dosages. The number after SC or SPSC is NaOH dosage.

**Figure 5 polymers-15-01693-f005:**
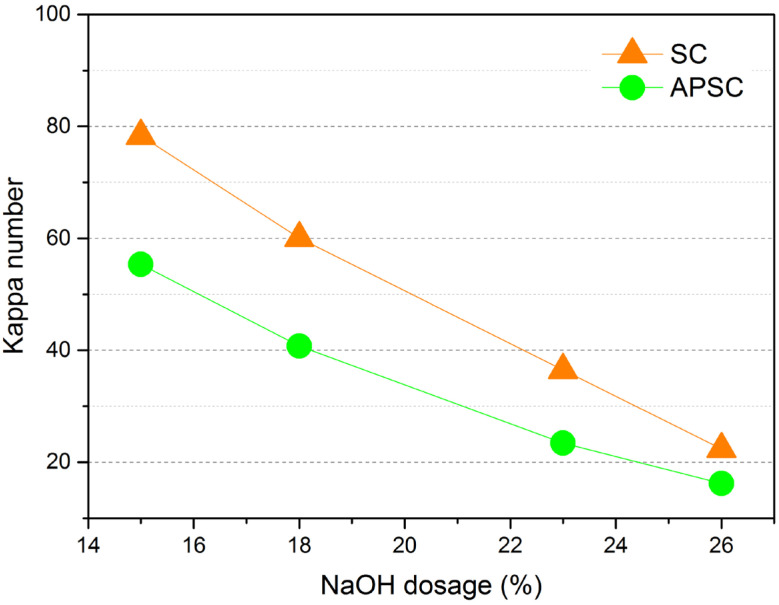
Comparison of Kappa number of screened pulp between SC and APSC at different NaOH dosages.

**Figure 6 polymers-15-01693-f006:**
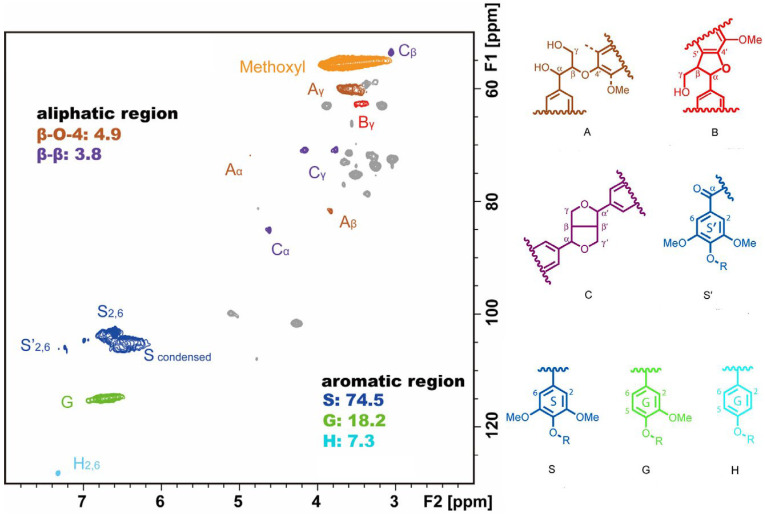
The 2D ^1^H-^13^C HSQC spectrum of APSC-AWW lignin, (A) β-O-4 aryl ether linkages with a free-OH at the γ-carbon; (B) phenylcoumarane substructures formed by β-5 and α-O-4 linkages; (C) resinol substructures formed by β-β, α-O-γ, and γ-O-α linkages; (S′) oxidized syringyl units with a C_α_ ketone or a C_α_ carboxyl group; (S) syringyl units; (G) guaiacyl units; (H) *p*-hydroxyphenyl units. The abundance of substructures was expressed per 100 aromatic units.

**Figure 7 polymers-15-01693-f007:**
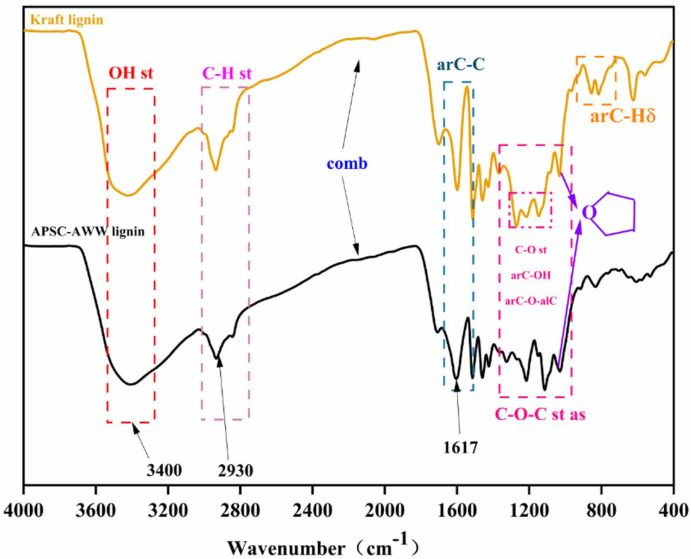
FTIR of APSC-AWW lignin and kraft lignin.

**Figure 8 polymers-15-01693-f008:**
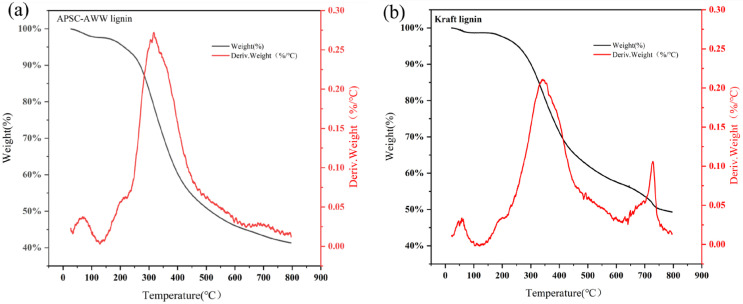
TGA of APSC-AWW lignin (**a**) and kraft lignin (**b**).

**Table 1 polymers-15-01693-t001:** Weight distribution of pulp fiber fractions retained on screens of 16, 30, 50, 100, 200 mesh (P_>16_, P_16–30_, P_30–50_, P_50–100_, P_100–200_) and passed through the screen of 200 mesh (P_<200_).

Pulp Samples *	Weight Distribution (%)					
	Long Fibers		Medium Fibers		Short Fibers	Fines
	P_>16_	P_16–30_	P_30–50_	P_50–100_	P_100–200_	P_<200_
APSC-AWW	0.9	8.4	45.2	22.1	16.5	6.9
KP-P	38.4	29.5	16.0	7.0	3.7	5.4
KP-EA	0.1	3.5	58.0	26.6	7.7	4.2
RP-OCC	16.9	17	19.1	11	6.6	29.4

* APSC-AWW: APSC pulp of AWW at 23% NaOH; KP-P: commercial unbleached kraft pulp of *Pinus massoniana*; KP-EA: commercial unbleached kraft pulp of mixed *Eucalyptus* and *Acacia Rachii*; RP-OCC: recycled pulp of old corrugated container.

**Table 2 polymers-15-01693-t002:** Comparison of physical properties of paper sheets from different biomass.

Pulp	Tensile Index (N·m/g)	Tear Index (mN·m^2^/g)	Burst Index (kPa·m^2^/g)	Reference
APSC-AWW	56.9	3.2	3.1	-
KP-P	75.8	14.3	5.0	-
KP-EA	56.4	6.1	4.6	-
RP-OCC	43.6	6.7	2.3	-
SC-CS ^a^	55.9	8.0	4.5	[21]
SC-WS ^b^	80.4	6.6	5.6	[22]
SC-PF ^c^	24.9	6.1	1.9	[23]
APSC-AWW/KP-P_(95/5)_ ^d^	58.2	4.6	3.6	-
APSC-AWW/KP-P_(90/10)_	60.5	5.1	3.8	-
APSC-AWW/KP-P_(80/20)_	64.6	6.8	4.3	-

^a^ SC-CS: unbleached soda pulp of corn stalk; ^b^ SC-WS: unbleached soda pulp of wheat straw. ^c^ SC-PF: unbleached soda pulp of palm fiber; ^d^ APSC-AWW/KP-P_(95/5)_: mixture of 95% APSC-AWW and 5% KP-P.

**Table 3 polymers-15-01693-t003:** Comparison of molecular weight among different lignins.

Lignin *	Mw (g/mol)	Mn (g/mol)	PDI (Mw/Mn)	Reference
APSC-AWW	4462	2032	2.2	-
Indulin kraft	4480	1100	4.1	[25]
Soda-P1000	3260	940	3.5	[25]
Soda-palm	3616	1915	1.9	[23]

* Indulin kraft is kraft lignin from softwood; Soda-P1000 is soda lignin from mixed wheat straw and Sarkanda grass; Soda-Palm is soda lignin from palm fiber.

## Data Availability

The data presented in this study are available on request from the corresponding author.

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
