# Peer review of "Production of Cellulose Pulp and Lignin from High-Density Apple Wood Waste by Preimpregnation-Assisted Soda Cooking"

_polymers, 2023, doi:10.3390/polym15071693_

Round 1

Reviewer 1 Report

The authors described the production of cellulose pulp and lignin from high-density apple wood waste by preimpregnation-assisted soda cooking. Although the subject is interesting, the importance of the research is unclear. The authors mention that “the conventional soda cooking was modified to overcome the hard and dense structure of AWW and therefore facilitate delignification.” Which is the relevance of the modification of conventional soda cooking? Is it the alkali preimpregnation the modification to the reported procedure? Why is it important to overcome the hard and dense structure of AWW? These are some questions that are not answered yet in the manuscript.

On the other hand, the authors mention pulp production as the most important application for the AWW, and it seems that the first chemical characterization of AWW was reported. However, nothing is mentioned related to the potential presence of heavy metals in AWW.

In Results and Discussion, the authors mentioned and discussed the ash content for AWW as 0.4%. However, it is unclear if this percentage is considered high or low, considering its influence on the high corrosion rates of pulping equipment. There are standard accepted ranges for ash contents? Which are the references?

In general, many phrases in the text are missing a hyphen, i.e. acid insoluble, well acknowledged, long fibered, please, revised.

Abstract

Line 10. Replace “truck” by “trucks”. It seems that truck may not agree in number with other words in this phrase.

Line 12. The noun phrase density seems to be missing a determiner before it. Consider adding “the” before “density”.

Line 17. Your sentence appears to use the incorrect form of Fibers. Consider changing it to singular “Fiber”.

Line 18. It appears that “the necessity of” may be unnecessary in this sentence. Consider removing it.

Line 18 and 19. It seems that “long fibered” is missing a hyphen. Consider adding the hyphen.

Line 22. Consider adding “a” before “weight-average”. Besides, replace “weigh” by “weight”.

Related to the sentence, “It is believe the utilization of fruit tree wood as feedstock of pulp industry is attractive to countries without sufficient forestry resources.” Is it really true? The use of fruit tree wood as feedstock of pulp industry should promote deforestation and, in the middle and long term, more serious environmental damage. Please, consider rewriting the sentence focuses on the apple wood waste.

Line 27. The noun phrase “feasibility” seems to be missing a determiner before it. Consider adding “the” before ““feasibility”.

Introduction

Line 33. Consider adding a comma before “which”.

Line 42. The phrase “the removal of” may be wordy. Consider changing the wording by “removing”.

Line 44. The phrase “be helpful in” may be wordy. Consider changing the wording by “help improve”.

Line 57. The to-infinitive to utilize has been split by the modifier more efficiently. Consider replacing “technologies to more efficiently utilize various low-cost” by “to utilize various low-cost  more efficiently”.

Line 59. It appears that the modifiers in the noun phrase “forest poor countries” are in the wrong order. Consider changing the word order by “poor forest countries”.

Line 64. The phrase “prior to” may be wordy. Consider changing by “before”.

Results and Discussion

Line 145. Consider using the plural form for “level”, replaced by “levels”.

Line 145. Consider replacing “lead” by “leads”.

Line 155. The noun phrase “pulping process” seems to be missing a determiner before it. Consider adding “the” before “pulping process”.

Line 188. Consider replacing the sentence “For comparison, fiber classification of KP-P, KP-EA, RP-OCC were also conducted” by “Fiber classification of KP-P, KP-EA, and RP-OCC were also conducted for comparison.

Line 196. It seems that the verb “is” does not agree with the subject. Consider changing the verb form by “are”.

Conclusions

AWW's chemical compositions before the exploration of cooking conditions were mentioned as one of the main goals of the research work; however, it is not included in the conclusions. Consider including a sentence related to this specific issue.

The research work will merit publication; however, you must consider all observations made.

Author Response

  1. Which is the relevance of the modification of conventional soda cooking?

Response: The modification is long time preimpregnation which enhances the permeation of cooking liquid into the wood of applewood.

  1. Is it the alkali preimpregnation the modification to the reported procedure?

Response: Yes, the conventional soda cooking is one step treatment. The modified process include alkali preimpregnation and soda cooking, which is two steps treatment.

3.Why is it important to overcome the hard and dense structure of AWW?

Response: Pulping is a defibering process. Preimpregnation and soda cooking can overcome the hard and dense structure of AWW by enhanced lignin removal and then facilitate defibering.

  1. The ash content for AWW as 0.4%. However, it is unclear if this percentage is considered high or low, considering its influence on the high corrosion rates of pulping equipment. There are standard accepted ranges for ash contents? Which are the references?

Response:

Ash content 0.4% is in the same lever to most wood bioamss used in pulping industry. Agriculture straws contains more minerals, and shows higher ash content, and not suitable for chemical plping.

Other major changes and responses to reviewer comments are presented in the appendix.

Reviewer 2 Report

The manuscript "Production of cellulose pulp and lignin from high-density apple wood waste by preimpregnation-assisted soda cooking" has the potential to be published after some modifications.

- the characterization of the extracted materials is very poor, making it difficult to compare them with the literature and with kraft materials; please insert at least one characterization that presents the morphology of the sample; a characterization of thermal degradation and FTIR. (if possible, also attach a DRX to link with the results obtained in the 2D 1H13C NMR.

Author Response

The manuscript "Production of cellulose pulp and lignin from high-density apple wood waste by preimpregnation-assisted soda cooking" has the potential to be published after some modifications.

- the characterization of the extracted materials is very poor, making it difficult to compare them with the literature and with kraft materials; please insert at least one characterization that presents the morphology of the sample; a characterization of thermal degradation and FTIR. (if possible, also attach a DRX to link with the results obtained in the 2D 1H13C NMR.

Response: Considering the Reviewer’s suggestion, we have added the characterization of the AWW materials in the conclusion and inserted the FTIR of APSC-AWW lignin and kraft lignin, the TGA of APSC-AWW lignin and kraft lignin. We also have inserted the Micro-CT Scanning of AWW and Pine.

Special thanks to you for your good comments.

Reviewer 3 Report

The experimental article "Production of cellulose pulp and lignin from high-density apple wood waste by preimpregnation-assisted soda cooking" meets Polymers according to the main criteria. The main hypothesis is reduced to the proposal to obtain pulp and lignin from superdense apple wood, while the technology is based on treatment with sodium hydroxide at a temperature of 160 °C. The authors carried out a rather reasonable control, but due to the high value of yield of rejects during the isolation of cellulose, one more stage was additionally introduced by preliminary impregnation with alkali at 90 °C (APSC). The strength of the article is not just the desire to turn non-target wood into useful products, but also the production of two products at the same time. The article is written in a fairly simple language, but at the same time it is obvious to the reviewer that the authors shift the main emphasis on the fact of obtaining lignin and the description of its properties, while it is possible to show much more results, specifically to bring the final data on the transformation of 1 kg of apple wood into so many then g of pulp - a component of composite paper, and so much g of lignin intended for a specific application. I myself chose this article for peer review because it reflects my scientific interests. The attempt of researchers to obtain valuable products from waste using simple processing steps, available reagents and equipment is attracting attention around the world. There is an active search for universal methods designed for a variety of sources of renewable raw materials or waste plant materials.

I think that the article should be finalized and then published.

Disadvantages to be fixed:

1. Since the chemical composition of apple wood is presented for the first time in the scientific literature, it is necessary to reflect this fact in the conclusion of the article.

2. It is necessary to cite MDPI articles that are similar in terms of the hypothesis of obtaining pulp for the paper industry from non-target types of raw materials. There is a similar material for guiding lignin.

3. If this publication is the only description of obtaining pulp from apple wood, then the authors should assume that the high proportion of short fiber characterizes the type of raw material, and not the method of extraction. There are no such considerations. While the authors describe the scope of the obtained cellulose only when mixed with long-fiber commercial celluloses.

4. There is no yield (from the mass of raw materials taken for processing) and the component composition of cellulose (APSC), obtained by a conditionally optimal method.

5. There is no yield (from the mass of raw materials taken into processing) of lignin obtained by a conditionally optimal method, and the main area of ​​its use. The main characteristic may not be the molecular weight, but rather the S / G / H ratio.

6. The authors do not discuss the need to use aqueous solutions of sodium hydroxide for the delignification of raw materials, but provide data on the S/G/H ratio of lignin. As a hypothesis, the authors were obliged to link the use of sodium hydroxide and the production of lignin with a certain S/G/H ratio.

7. The annotation is very well written and the poor conclusion is unacceptable. Expand the conclusion.

Author Response

Disadvantages to be fixed:

  1. Since the chemical composition of apple wood is presented for the first time in the scientific literature, it is necessary to reflect this fact in the conclusion of the article.

Response: Thank you. We added in the abstract ‘In present work, chemical compostion determiantion and microstructure observation was performed for the first time to AWW’.

  1. It is necessary to cite MDPI articles that are similar in terms of the hypothesis of obtaining pulp for the paper industry from non-target types of raw materials. There is a similar material for guiding lignin.

Response: Yes, references 11 and 19 are from MDPI JOURNAL (POLYMERS). we cited MORE MDPI articles about birefinery in the revised version,references 7 and 18.

  1. If this publication is the only description of obtaining pulp from apple wood, then the authors should assume that the high proportion of short fiber characterizes the type of raw material, and not the method of extraction. There are no such considerations. While the authors describe the scope of the obtained cellulose only when mixed with long-fiber commercial celluloses.

Response: Yes, from our reseach, we conclude apple wood is charactrerized with short firber. The developed process is capable of seperating these short fibers from apple wood. Conventional soda pulping can not do this.

  1. There is no yield (from the mass of raw materials taken for processing) and the component composition of cellulose (APSC), obtained by a conditionally optimal method.

Response: Please see figure 4. There are value of yields of screened pulp.  ‘Figure 4 Pulping performance of SC and APSC in terms of yield of screened pulp and yield of rejects at different NaOH dosages.‘

From Figure 4 and Figure 5, the screened pulp of APSC-26 showed a yield of 35.5% and a Kappa number of 16.2. we think conditions of APSC-26 is OK.

  1. There is no yield (from the mass of raw materials taken into processing) of lignin obtained by a conditionally optimal method, and the main area of ​​its use. The main characteristic may not be the molecular weight, but rather the S / G / H ratio.

Response: Line 268-270 “Lignins with higher Mw and lower PDI is preferred for dye dispersant production due to its outstanding interface performance and thermostability.” introduces the use area of lignin.

  1. The authors do not discuss the need to use aqueous solutions of sodium hydroxide for the delignification of raw materials, but provide data on the S/G/H ratio of lignin. As a hypothesis, the authors were obliged to link the use of sodium hydroxide and the production of lignin with a certain S/G/H ratio.

Response: THIS IS A VERY GOOD QUESITON. Generally speaking, G unit is dominant in softwood lignin, which is the resson why sodium sulphide is need for softwood delignification during kraft pulping process. We added this explaination in the revised manuscript.

  1. The annotation is very well written and the poor conclusion is unacceptable. Expand the conclusion.

Response:

Special thanks to you for your good comments. We rewrited conclusion.

The detailed changes are presented in the attachment.

Round 2

Reviewer 2 Report

The manuscript improved its quality substantially during the review process. I am happy to recommend it for publication in the Polymers Journal.